# The Relationship between Mindfulness Practices and the Psychological State and Performance of Kyokushin Karate Athletes

**DOI:** 10.3390/ijerph19074001

**Published:** 2022-03-28

**Authors:** Jolita Vveinhardt, Magdalena Kaspare

**Affiliations:** 1Institute of Sport Science and Innovations, Lithuanian Sports University, 44221 Kaunas, Lithuania; 2Department of Sport and Tourism Management, Lithuanian Sports University, 44221 Kaunas, Lithuania; magdalena.kaspare@gmail.com

**Keywords:** mindfulness, stress management, psychological state, emotional state, performance, kyokushin karate, sport

## Abstract

The aim of this study was to determine the relationship between mindfulness practices and the psychological state and qualification of kyokushin karate athletes. The survey was conducted using the Mindful Attention Awareness Scale (MAAS-15) and the Depression, Anxiety and Stress Scale (DASS-21). The study involved 371 Lithuanian kyokushin karate athletes (of which 59.3% were male and 40.7% were female; 71.4% of research participants have practiced this sport for 11 and more years and have the 1st dan or a higher belt). The results of the study showed a positive impact of mindfulness in reducing stress experienced by athletes, improving their psychological state, and enhancing their athletic performance. A moderate negative correlation was identified between stress, anxiety, and mindfulness, and while the mindfulness score was increasing, the severity level of depression was decreasing. Meanwhile, the correlation of the meditation effect and anxiety with kyokushin karate 0–7 kyu belt was very weak but statistically significant. The research results could be useful not only for athletes and their coaches but also for sports organizations. After analysing the benefits of mindfulness for kyokushin karate athletes, mindfulness practices are proposed for the effective improvement of athletes’ physical and psychological state when preparing for professional-level competitions.

## 1. Introduction

Stress in professional sports is often experienced during training and competitions as well as during the competition period. According to research conducted by Borkoles et al. (2018) [1], competitive sport is always associated with stressful experiences. Garinger et al. (2018) [2] argue that athletes experience a host of stressors, including daily worries, conflicts, relationship problems, and training difficulties. If stress is not managed effectively, it can intensify over time and, in the long run, can have negative consequences for the athlete. One of the most popular stress management measures is mindfulness. Mindfulness is a practice related to various religious and secular traditions ranging from Hinduism and Buddhism to yoga and nonreligious meditation that have only recently been introduced [3]. Mindfulness is increasingly being used in educational institutions around the world (e.g., in the USA, the UK, Iceland, and the Netherlands) through programs such as “Inner Kids”, “MindUp”, and “MindfulKids”. The incorporation of mindfulness interventions in education is motivated not only by the desire to improve universal mental health but also by the hope to see better academic outcomes [4]. According to Josefsson et al. (2019) [5], the focus on external stimuli (rather than internal experiences) increases the ability to accurately process the latest information. Mindfulness and information perception skills can reduce reactivity, improve performance at cognitive tasks, and reduce stress. Mindfulness affects cognitive processes and is increasingly considered a meaningful approach for psychological training [6]. Mindfulness practices are used by athletes of various sports branches, including kyokushin karate. However, no research has been done on the impact of these practices on the emotional, psychological, and physical states of athletes, namely those doing kyokushin karate, and on the effect of these practices on their athletic performance. Practices that would ensure athletes’ optimal psychological state are a significant research area addressed by sports scientists. Athletes experience physical and psychological trials, and preparation for sports competitions alone promotes feelings and thoughts associated with risk and the fear of failure and may have a negative impact on sports performance [5,7]. In order to improve athletes’ mental health and relieve tension in high-performance sports, it is proposed that mindfulness programs should be introduced [5,8,9,10,11]. Research shows that this practice positively influences athletes’ mood, level of awareness, anxiety, emotional state, and ability to cope with stress [4,12,13]. In addition, mindfulness is used to prevent injuries, reduce burnout, enhance self-confidence [14], and improve athletic performance [8]. Still other studies have shown that mindfulness can be beneficial for athletes of various ages [6,13], and the positive effect has been confirmed in different sports [8].

In recent years, several studies were conducted, investigating the benefits of mindfulness in karate, a highly dynamic sport demanding effort and self-control, accompanied by high stress levels [13,15]. For example, Jansen et al. (2017) [13] investigated how mindfulness-based stress reduction intervention was related to cognitive functioning and personal well-being in older adults. Although the overall effect was small, the authors believe that mindfulness-based stress reduction intervention may be a useful measure. Miyata et al. (2020) [16] investigated the question of whether martial arts practice based on Japanese traditions (combining physical and mental practices) was related to mindfulness and psychological health. It was found that the use of these techniques was related to a better psychological state, and the longer and more frequent the practices were, the better the state was. A positive effect on the psychological state was found while analysing various styles: aikido [17], kung fu [18], and taekwondo [19]. It should be noted, however, that in the context of this sport, the kyokushin style stands out; according to Piepiora and Petecka (2020) [20], it is one of the most difficult, has a knockdown system, and competition generates stress. Although various psychoregulatory techniques are used for stress management, allowing one to achieve an optimal psychological state during combat to concentrate the attention [21], so far, there is a lack of data on how the use of mindfulness practices pertains not only to better psychological state but also to athletes’ mastery.

Kyokushin karate. The kyokushin karate style, founded by the Japanese master Masutatsu Oyama, originated in the middle of the 20th century. Like other styles of karate, kyokushin includes three components: kihon (basic blocks and blows), kata (a model of blocks/attacks against imaginary opponents), and kumite (fighting against real opponents). Kyokushin karate distinguishes itself by meditations and teaching according to the moral principles of martial arts [22]. This style is considered to be one of the most difficult karate styles; it has a knockdown system [20], and athletes do not always use protective equipment and can suffer significant damage [23]. Karate distinguishes itself by fast legwork, sudden kicks of the lower limbs, different techniques, ad changes in distance [24]. The fight takes place between two fighters without any weapons but at maximum speed and strength [25], while the arm and leg strokes dominate to score points or win the fight [26]. The fighter’s mastery, which depends not only on physical but also on psychological preparation, is demonstrated by the kyu and dan system [22,27,28].

Mindfulness and meditation. Mahmood et al. (2018) [29] state that mindfulness is one of the new techniques considered in sports psychology to improve athletes’ performance. Mindfulness is characterized by the ability to direct attention to the present moment, recognize emerging experiences, and accept them [30,31,32,33]. External events are what the person cannot predict and control [34], and the cultivation of mindfulness helps to perceive one’s mood, respond to surrounding stimuli more effectively, and reduce stress [31]. Mindfulness is inseparable from meditation, which encompasses a nonjudgemental perception of the present moment, and the goal of this practice is to develop a stable and nonreactive perception of one’s internal (cognitive–affective–sensory) and external (social–environmental) experiences [35]. At the same time, meditation contributes to the person’s growth. Nilsson and Kazemi (2016) [34], who have examined several dozen concepts of mindfulness, note that, in the Western understanding, cultivation refers to the nurturing or development of character through awareness that provides resilience to adverse events. In the Buddhist understanding, meanwhile, cultivation is associated with meditation that consists of calm and insight.

The athlete’s psychological state. Although intense sporting activities often lead to psychological well-being [36,37], competitive sport is always associated with stressful experiences [2,38]. During training, stress is experienced due to internal stimuli, mood swings, and health condition. During competitions, stress is experienced due to psychological tension (the desire to win or to be the best, fear of injury, etc.) [1]. All of this is related to increased cortisol levels, anxiety, and depression [2,39,40]. Meta-analyses have shown that 34% of elite athletes suffer from anxiety/depression (cf., 26% of former elite athletes experienced these symptoms). Meanwhile, Rice et al. (2019) have found that the severity of anxiety experienced by athletes is similar to that experienced by nonathletes [41]. Similar trends were found with regard to reports about mild or severe depression symptoms. In other words, athletes are not more protected from anxiety and depression because of their activities.

The aim of this study is to determine the relationship between mindfulness practices and the psychological state and performance of kyokushin karate athletes.

## 2. Materials and Methods

The empirical study was conducted using a questionnaire survey method. The survey employed the Mindful Attention Awareness Scale—15 items (MAAS-15) [42,43] and the Depression, Anxiety and Stress Scale—21 items (DASS-21) [44] as well as questions to determine the emotional state and demographic questions developed by the authors. Based on the theoretical analysis of the research problem, gender, age, duration of sporting activity, experience with kyokushin karate, and intensity of sporting activity were selected as demographic characteristics that may have links with motivation to engage in sports and improve performance as well as with the athlete’s psychological state. The questionnaire was prepared by translating the MAAS-15 and DASS-21 scales into Lithuanian. All procedures required for linguistic and cultural adaptation were followed while adapting the questionnaire [45]. The translations into Lithuanian were performed by two independent translators who speak Lithuanian and English and were unfamiliar to each other. Then, back translation was performed, and the translation was compared to the original. The semantics and content of the statements were found to be consistent with the original. The study was approved by the ethics committee of the Lithuanian Sports University, protocol No. SMTEK-26. Data were collected from March to June of 2021 using Google Forms survey administration software. The effect of meditation was analysed by Q6 questions. If the respondent answered one of Q6.1–Q6.3 positively, it meant that the meditations had had an effect.

The statistical significance of data differences was determined according to the χ^2^ criterion. Pearson correlation was used to determine the relationships of normal distributions, Spearman correlation was employed for ordinal variables, and coefficients of determination (*R^2^*) were assessed to find the effect of the latent factor on the observed variable. The following eligibility criteria were evaluated in the structural models: χ^2^, the Tucker–Lewis index (TLI), root mean square error of approximation (RMSEA with 90% confidence intervals), the comparative fit index (CFI), and the normed fit index (CMIN/DF). The Depression, Anxiety, and Stress Scale (DASS-21) was classified into severity levels according to S. H. Lovibond, P. F. Lovibond, 1995 [44]. The level of statistical significance *p* < 0.05. Statistical analyses were performed using IBM SPSS 26.0 and IBM SPSS AMOS 27 software (IBM—1 New Orchard Road Armonk, NY 10504-1722, USA).

## 3. Results

### 3.1. Organization of the Research

The study was conducted by employing a questionnaire, as we aimed to collect as many responses from professional kyokushin athletes as possible. Lithuanian kyokushin athletes live in different cities and even countries; therefore, the questionnaire was the most effective and efficient way to obtain accurate data. The quantitative type of research was chosen. According to data from the Lithuanian Kyokushin Karate Federation in 2016, there are 4925 adults engaged in the sport in Lithuania. The appropriate sample size was calculated using the Paniotto equation: *n* = 1/ (Δ2 + 1/*N*), where: *n* is the number of individualsin the sample; Δ is the error value; and *N* is the total population, 4925. Using the equation with a sample error of 5% and a level of confidence of 95% (*n* = 1/((0.05) 2 + 1/4925), we found that the sample size for this study must be at least 370 individuals. Respondents were selected by systematic random sampling. A certain number of athletes from each Lithuanian kyokushin club were randomly surveyed. That is, every tenth athlete was selected from the athlete lists provided by the clubs, and then consent was sought. Consent was obtained from 382 individuals. A link to the questionnaire was sent to those who gave their consent to the e-mail address provided by them. The questionnaire was not completed by 11 athletes.

### 3.2. Descriptive Data Analysis

The research sample consisted of 371 respondents, 59.3% male and 40.7% female, whose age ranged from 18 to 56. The age of the respondents ranged from the late twenties to mid-thirties (M = 27.75 years, SD = 8.56); 50.4% had a higher university education while 21.8% had secondary, 15.1% upper secondary, 9.4% higher nonuniversity education, and 1.3% only primary education, and 1.9% special secondary education. Almost 75% of respondents were Christians and 97% lived in Lithuania.53.6% of all respondents had practiced kyokushin for 11 years or more (); 14.8% had practiced this sport for up to five years, and 31.5% for 6–10 years. The more years the athlete had been practicing karate, the higher-ranked belt he or she had (χ^2^ (4) = 240.9, *p* < 0.001); 71.4% of athletes who have been practicing the sport for 11 years or more have a 1st dan or higher belt (Table 1).

### 3.3. The Effect of Meditations and Mindfulness

According to the survey results, there were 84.4% meditating athletes and 15.6% nonmeditating athletes. Based on the MASS-15 rating scale, variants in all responses were summed up. The higher the score obtained, the more effective the mindfulness practice. We observed that mindfulness is statistically significantly related to the severity of anxiety, stress, and depression. There is a negative correlation, of moderate strength, between signs of stress, anxiety, and mindfulness: as the score for mindfulness increases, the level of stress and anxiety decreases. There is a statistically significantly negative, but weak, correlation between signs of depression and mindfulness, but, likewise, as the mindfulness score increases, the severity level of depression decreases (Table 2). 

### 3.4. Structural Factor Models by Meditation Impact Groups

The RMSEA index indicates a fairly good fit between the model and the data, where the 90% confidence interval at the lower end is less than 0.05 and at the upper end it is less than 0.10. The main statistics are as follows: χ^2^ (1604) = 2957.88, *p* < 0.001, probability < 0.001, CFI = 0.848, TLI = 0.836, CMIN/DF = 1.844.

The emotional state of meditating athletes is explained most by “the feeling of unfounded anxiety” (Q7A3); all observable variables are statistically significantly related to the emotional state (a *p*-value less than < 0.001) and are suitable for predicting emotional state. The emotional state of meditating athletes included “feeling unloved” (Q7A2) the least; the emotional state of only 9.5% included “feeling lonely” (QA71), while “experiencing strong stress” (QA74), or “distraction” (Q7A5) fluctuated from 31% to 36%, with a 61.6% variance competition explained by the impact of meditation. Nonmeditating athletes who found it difficult to calm down tended to react poorly to situations, felt that they were using a lot of nervous energy, found themselves getting agitated, found it difficult to relax, were intolerant of everything, and felt rather touchy (Q9A1–Q9A7; Q9A1 *R^2^* = 0.662, *p* < 0.05, Q9A2 *R^2^* = 0.440, *p* < 0.05, Q9A3 *R^2^* = 0.512, *p* < 0.05, Q9A4 *R^2^* = 0.363, *p* < 0.05, Q9A5 *R^2^* = 0.700, *p* < 0.05, Q9A6 *R^2^* = 0.382, *p* < 0.05, Q9A7 *R^2^* = 0.510, *p* < 0.05). In meditating athletes, signs of stress explained 64.5% of the variance in energy loss while nervous. The coefficients of determination (*R^2^*) were large for all observable variables and are therefore explained by a better percentage. Dry mouth, breathing difficulties, trembling, worrying about situations in which one might panic and make a fool of oneself, having nothing to look forward to, feeling downhearted and blue, and an absence of enthusiasm were statistically significantly related to signs of anxiety (*p* > 0.001). Anxiety is the main explanation for panicking. Signs of depression are statistically significantly related to all observable variables, with depression often being described as “having nothing to look forward to” (Q11A3). All observable variables had high coefficients of determination (*R^2^*), except for “working up the initiative to do things” (Q11A2). Most factors had a strong or moderate correlation; the emotional state is highly dependent on situations, but has negative relationship with feelings, state, and how one reacts to certain situations. The emotional state of meditating and nonmeditating athletes is explained by “recently experienced unfounded anxiety”, only of different intensity—52% and 31.7%, respectively. The emotional state is statistically significantly related to “recently feeling unloved” among nonmeditating athletes, but when comparing groups, we can see that the latter observable indicator is least explained by meditating athletes’ emotional state. Not all observable variables of mindfulness in the model were statistically significant explaining factors, e.g., “the tendency to walk quickly to get where one is going without paying attention to what one experiences along the way” (Q8A4) and “forgetting a person’s name almost as soon as one has been told it for the first time” (Q8A6) are statistically significantly unrelated (*p*-value greater than 0.05); 49.1% of the variance in “finding oneself preoccupied with the future or the past” (Q8A13) depended on mindfulness, and in the group of meditating athletes, such an observable variable was “running on automatic, without much awareness of what one was doing” (Q8A7). Composite reliability was more than 0.6 for signs of depression, emotional state, and mindfulness; for signs of stress and anxiety it was more than 0.7, so they are a good fit, as Fornell and Larcker (1981) [46] recommended a CR value of 0.60 or more (Table 3).

In the structural factor model, exogenous (latent) factors associate and have estimates. Emotional state has a strong correlation with mindfulness. When mindfulness is higher, athletes’ emotional state is better. Signs of stress moderately correlate with signs of anxiety and depression. The higher the stress level, the more severe the signs of anxiety and depression. Mindfulness has a moderate negative correlation with signs of stress, anxiety, and depression. As mindfulness increases, indicators of stress, anxiety, and depression decrease (Table 4).

### 3.5. Links between Mindfulness and Athletes’ Performance 

The exogenous variable—the kyokushin belt and other observable indices and latent factors (the impact of meditation, emotional state, mindfulness, signs of stress, anxiety, depression) were observed in the model (CFI = 0.895; TLI = 0.89, RMSEA = 0.07, CI 90 = 0.067–0.074). All observable variables were statistically significantly related to latent factors (*p*-value < 0.001). Based on the coefficients of determination (*R^2^*), it was observed that the models constructed for different variables explain a different proportion of the variance of dependent variables. Based on this, it has been found that 61.6% of the variance in variable meditation explains that meditation before competition helps athletes to calm internal stimuli in this model.

There was a positive influence of meditation on athletic performance. Signs of stress were statistically significantly related to all observable variables (*p*-value < 0.001). Signs of stress explained 50.3% of variance in “finding it hard to wind down”; 52.1% of “the tendency to over-react to situations”; 61.8% of “feeling that one was using a lot of nervous energy”; 47.9% of “finding oneself getting agitated”; 58.8% of “finding it difficult to relax”; and 47.5% of “being intolerant of anything that kept one from getting on with what one was doing”. High coefficients of determination describe the model well and are suitable for use. Signs of anxiety were statistically significantly related to all observable variables (*p*-values < 0.001). Signs of anxiety explained 62.4% of variance in “athletes’ worrying about situations in which they might panic and make a fool of themselves” but explained “dryness of mouth” least (only 33.1% of variance). Signs of depression explain the different proportions of variance of dependent variables. Signs of depression most explained athletes’ “feeling of being down-hearted and blue”, with 64% of variance. They least explained athletes’ “difficulty working up the initiative to do things”. Signs of depression were statistically significantly related to all observable variables (*p*-value < 0.001).

## 4. Conclusions

After establishing connections between mindfulness practice and the psychological state and performance of kyokushin karate athletes, we can conclude that mindfulness practices are related to athletes’ higher performance and better psycho-emotional state. After performing a confirmatory factor analysis, we observed from the data that the psychological state indicators of kyokushin athletes who used mindfulness techniques (meditation, concentration) were better than those of athletes who did not use these techniques and were unaware of the benefits of mindfulness. Hence, mindfulness practices have a positive impact on athletes’ psycho-emotional state and help them to better prepare for competitions. When the research model was supplemented with the exogenous variable, i.e., the athlete’s performance level, as reflected by the kyokushin karate belt possessed, it was found that better psychological indicators were related to the athlete’s performance. As mindfulness practices improve athletes’ health indicators, their athletic performance improves, too. The benefits of mindfulness for athletes are observed not only in dual sports but also in team and individual sports. Our research shows that, while kyokushin athletes are in training, it makes sense to incorporate mindfulness practice for several reasons. Athletes’ health indicators improve due to mindfulness, and their likelihood of better performance in the sport also increases. Although the training of kyokushin karate athletes should be inseparable from meditation, our research has shown that athletes use meditation. To find out why athletes do not use meditation requires further research in the future, but it must also be taken into account that mindfulness is more than just meditation techniques. In addition to the fact that mindfulness reduces anxiety, depression, and stress levels, these practices would also be useful for improving athletes’ attention.

## Figures and Tables

**Table 1 ijerph-19-04001-t001:** Characteristics of respondents.

Characteristic		Male *N* (%)	Female *N* (%)	Total within Groups *N* (%)
		220 (59.3)	151 (40.7)	371 (100)
Kyokushin belt classification	0–7 kyu	14 (6.4)	14 (9.3)	28 (7.5)
	6–1 kyu	100 (45.5)	74 (49.0)	174 (46.9)
	1st dan or higher	106 (62. 7)	63 (37.3)	169 (45.5)
Years of practicing kyokushin	less than 5 years	30 (13.6)	25 (16.6)	55 (14.8)
	6–10 years	67 (30.5)	50 (33.1)	117 (31.5)
	11 years or more	123 (55.9)	76 (50.3)	199 (53.6)

Notes: Kyu means “class” in Japanese. When a pupil begins attending karate classes, he or she is given a white belt that is not classified. Athletes take the first exam to get an orange belt, which is classified as “10 kyu”. The higher the belt exam, the lower the number of “kyu”. Dan means the black belt class in Japanese. The first black belt is classified as “1 dan”. The higher the belt exam the athlete takes, the higher the number of “dan”. It is important to mention that the number of the athlete’s “dan” corresponds to the number of gold stripes on his black belt.

**Table 2 ijerph-19-04001-t002:** Relationship between mindfulness and the severity of stress, anxiety, and depression.

Subscales *N* = 371 (M, SD)	Normal *N* (%; M; SD)	Mild *N* (%; M; SD)	Moderate *N* (%; M; SD)	Severe *N* (%; M; SD)	Very Severe *N* (%; M; SD)	χ^2^ (df), *p*-Value, Correlation
Stress (13.53; 4.16)	220 (59.3; 10,75; 2,37)	103 (27.8; 16.05; 1.05)	47 (12.7; 20.66; 0.71)	1 (0.3; 27.0; 0.0)	0 (0.0)	χ^2^ (195) = 604.94, *p* < 0.001. r = −0.514 *
Anxiety (10.71; 3.56)	76 (20.5; 7.0; 0.0)	92 (24.8; 8.52; 0.50)	154 (41.5; 11.62; 1.44)	38 (10.2; 16.47; 1.31)	11 (3; 22.00; 3,19)	χ^2^ (260) = 370.04, *p* < 0.001. r = −0.605 *
Depression (11.19; 4.11)	157 (42.3; 7.92; 0.80)	192 (51.8; 12.55; 2.38)	4 (1.1) 20	16 (4.3; 22.75; 1.13)	2 (0.05; 28.0; 0.0)	χ^2^ (260) = 459.26, *p* < 0.001. r = −0.468 *

Abbreviations: *N*—cases, M—mean, SD—standard deviation. * *p*-value < 0.001.

**Table 3 ijerph-19-04001-t003:** Nonstandardized and standardized estimates for the analysis of observable variables in meditating athletes (*N* = 313).

Factors	Variables	Nonstandardized					Standardized		Composite Reliability (CR)	Average Variance Extracted (AVE)
		B	95 CI (Lower, Upper)	SE	CR	*p*-Value	β	*R^2^*		
Emotional state	Felt lonely (Q7A1)	1.000	1.00–1.00				0.497	0.247	0.643	0.579
	Felt unloved (Q7A2)	0.459	0.196–0.665	0.082	5.614	<0.001	0.308	0.095		
	Felt unfounded anxiety (Q7A3)	1.745	1.341–2.727	0.223	7.810	<0.001	0.718	0.516		
	Experienced strong stress (Q7A4)	1.659	1.239–2.720	0.231	7.179	<0.001	0.604	0.364		
	Felt distracted (Q7A5)	1.637	1.207–2.603	0.232	7.044	<0.001	0.583	0.340		
	Felt unable to focus on one activity (Q7A6)	1.561	1.124–2.544	0.227	6.884	<0.001	0.558	0.311		
Mindfulness	I could be experiencing some emotion and not be conscious of it until some time later (Q8A1)	1.000	1.00–1.00				0.784	0.615	0.875	0.501
	I break or spill things because of carelessness, not paying attention, or thinking of something else (Q8A2)	0.848	0.756–0.971	0.069	12.361	<0.001	0.663	0.439		
	I find it difficult to stay focused on what’s happening in the present (Q8A3)	0.858	0.64–0.961	0.064	13.354	<0.001	0.706	0.499		
	I tend to walk quickly to get where I’m going without paying attention to what I experience along the way (Q8A4)	1.047	1.002–1.182	0.085	12.379	<0.001	0.663	0.364		
	I tend not to notice feelings of physical tension or discomfort until they really grab my attention (Q8A5)	0.896	0.769–0.989	0.087	10.259	<0.05	0.564	0.439		
	I forget a person’s name almost as soon as I’ve been told it for the first time (Q8A6)	1.054	1.001–1.121	0.086	12.201	<0.001	0.655	0.311		
	It seems I am “running on automatic”, without much awareness of what I’m doing (Q8A7)	1.041	1.010–1.210	0.068	15.258	<0.001	0.788	0.318		
	I rush through activities without being really attentive to them (Q8A8)	1.016	0.894–1.56	0.065	15.514	<0.05	0.796	0.429		
	I get so focused on the goal I want to achieve that I lose touch with what I’m doing right now to get there (Q8A9)	1.115	0.993–1.289	0.074	15.134	<0.05	0.781	0.620		
	I do jobs or tasks automatically, without being aware of what I’m doing (Q8A10)	0.926	0.784–1.109	0.071	13.100	<0.05	0.774	0.634		
	I find myself listening to someone with one ear, doing something else at the same time (Q8A11)	1.043	0.906–1.207	0.080	12.991	<.05	0.696	0.476		
	I drive places on ‘automatic pilot’ and then wonder why I went there (Q8A12)	0.955	0.821–1.168	0.073	13.123	<0.001	0.709	0.484		
	I find myself preoccupied with the future or the past (Q8A13)	1.109	1.012–1.267	0.083	13.414	<0.001	0.745	0.502		
	I find myself doing things without paying attention (Q8A14)	0.920	0.811–1.051	0.065	14.264	<0.05	0.626	0.555		
	I snack without being aware that I’m eating (Q8A15)	0.869	0.749–1.051	0.075	11.567	<0.001	0.796	0.392		
Signs of stress	I found it hard to wind down (Q9A1)	0.889		0.088	10.068	<0.001	0.703	0.494	0.904	0.575
	I tended to over-react to situations (Q9A2)	1.282	1.051–1.510	0.110	10.127	<0.05	0.709	0.502		
	I felt that I was using a lot of nervous energy (Q9A3)	1.395	1.162–1.588	0.126	11.034	<0.001	0.803	0.645		
	I found myself getting agitated (Q9A4)	1.058	0.913–1.331	0.104	10.172	<0.05	0.713	0.508		
	I found it difficult to relax (Q9A5)	1.114	1.015–1.331	0.110	10.127	<0.001	0.756	0.572		
	I was intolerant of anything that kept me from getting on with what I was doing (Q9A6)	1.160	0.967–1.221	0.115	10.091	<0.05	0.705	0.497		
	I felt that I was rather touchy (Q9A7)	1.000	1.00–1.00				0.605	0.366		
Signs of anxiety	I was aware of dryness in my mouth (Q10A1)	0.689	0.496–938	0.079	10.384	<0.05	0.556	0.309	0.784	0.523
	I experienced difficulty breathing (e.g., excessively rapid breathing, breathlessness in the absence of physical exertion) (Q10A2)	0.639	0.450–858	0.070	9.184	<0.05	0.556	0.309		
	I experienced trembling (e.g., in the hands) (Q10A3)	0.825	0.607–1.129	0.079	10.384	<0.05	0.630	0.397		
	I was worried about situations in which I might panic and make a fool of myself (Q10A4)	1.141	0.946–1.383	0.096	11.938	<0.05	0.728	0.530		
	I felt I was close to panic (Q10A5)	0.990	0.794–1.211	0.079	12.577	<0.05	0.769	0.591		
	I was aware of the action of my heart in the absence of physical exertion (e.g., sense of heart rate increase, heart missing a beat) (Q10A6)	0.937	0.750–1.187	0.086	10.949	<0.05	0.665	0.442		
	I felt scared without any good reason (Q10A7)	1.000	1.000–1.000				0.709	0.502		
Signs of depression	I couldn’t seem to experience any positive feeling at all (Q11A1)	1.078	0.884–1.307	0.077	13.962	<0.05	0.755	0.570	0.682	0.631
	I found it difficult to work up the initiative to do things (Q11A2)	0.847	0.627–1.160	0.084	10.106	<0.05	0.571	0.326		
	I felt that I had nothing to look forward to (Q11A3)	1.214	1.028–1.516	0.076	16.021	<0.05	0.858	0.735		
	I felt downhearted and blue (Q11A4)	1.103	0.947–1.431	0.075	14.651	<0.05	0.795	0.631		
	I was unable to become enthusiastic about anything (Q11A5)	1.078	0.834–1.348	0.077	13.962	<0.05	0.762	0.581		
	I felt I wasn’t worth much as a person (Q11A6)	1.099	0.884–1.307	0.076	14.524	<0.05	0.789	0.622		
	I felt that life was meaningless (Q11A7)	1.000	1.000–1.000				0.762	0.580		

Abbreviations: B—unstandardized estimate, 95 CI—95% confidence intervals, SE—standard error, CR—critical ratio, β—standardized regression weight, *R^2^*—R-squared.

**Table 4 ijerph-19-04001-t004:** Estimates of the latent factors (meditating athletes) in *structural factor model* (*N* = 313), χ^2^ (1604) = 2957.88, *p* < 0.001.

Factors	Estimate	SE	CR	*p*-Value	r
Emotional state ⇔ Mindfulness	0.11	0.019	5.837	0.001	0.590
Emotional state ⇔ Signs of stress	−0.061	0.010	−5.899	0.001	−0.739
Mindfulness ⇔ Signs of stress	−0.400	0.054	−7.401	0.001	−0.723
Emotional state ⇔ Signs of anxiety	−0.062	0.01	−6.038	0.001	−0.701
Signs of stress ⇔ Signs of anxiety	0.214	0.029	7.396	0.001	0.822
Mindfulness ⇔ Signs of anxiety	0.214	0.029	7.396	0.001	0.604
Signs of depression ⇔ Emotional state	−0.067	0.011	−6.387	0.001	−0.760
Signs of depression ⇔ Mindfulness	−0.346	0.047	−7.403	0.001	−0.587
Signs of depression ⇔ Signs of stress	0.180	0.025	7.160	0.001	0.690
Signs of depression ⇔ Signs of anxiety	0.208	0.026	7.953	0.001	0.750

Abbreviations: estimate—covariance, SE—standard error, CR—critical ratio, r—correlation.

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
