# Peer review of "The Relationship between Mindfulness Practices and the Psychological State and Performance of Kyokushin Karate Athletes"

_ijerph, 2022, doi:10.3390/ijerph19074001_

Round 1

Reviewer 1 Report

Previous observation: meditation is part of karate' tehniques learning, namely through katas' breathing, but this important variable was not analyzed, in either groups.

lines 46-48- review ponctuation "...with the fear of failure, risk 
and may have..."

lines 46, 58, 70- avoid repetition of the word "huge"

lines- 108-110- identify reference of meta-analysis

lines 112-113- not explicit which reference

lines 125-126- need to prove questionnaire cultural adaptation and validation 

lines 126-127- include process number

line 144- use APA norms 

lines 142-151- present statistical information in a table

lines 152-153- repeated information

lines 153-155- explore and present statistical association

Materials & Methods:
- need to prove consent

- need to present plataform used for data collecting

- nedd to present period of data collection

- need to present "Statistical Treatment"

lines 156-157- it is not explicit how meditation practice was defined and obtained (also Methods)

table 1

- statistical techniques used must be explicitly identified (Methods) and confidence intervals of association must be presented

- APA norms must be applied

- mean and sd values of  severity levels of stress, anxiety, depression signs must be presented

lines 178-180- no statistical information is presented for non-meditation group

Table 3. Correlations between latent factors (meditating athletes):

- Identify statistical technique, degrees of freedom, present probability and confidence intervals

lines 181-196- it is not clear if these data belong to non-meditating group, reorganize text to clarify it

Table 2. Non-standardized and standardized estimates for the analysis of observable variables in meditating athletes- statistical symbols must be identified somewhere... 

Links between mindfulness and athletes’ qualification- clearly clarify if it is the whole sample or not...

lines 244-245- for general public do not know system of belts classification, so will not understand significance of these specific results, namely, because there is and inverse significance between number and experience in kyu bekts and dan belts 

Table 4. Correlations between latent factors and observable exogenous variable- again, identify statistical technique, degrees of freedom, present probability and confidence intervals

lines 267-270- it seems that a linear correlation is not the adequate one, maybe quadratic? inverted U shape? and if so, what the consequences for discussion?

line 277- with so low correlation you need to present confidence intervals, and, once again, you need to identify what statistical technique was used

lines 286-288- identify that they were better in...

lines 288-290- speculative, your experimental design does not allow you to conclude this

lines 299-303- speculative, your experimental design does not allow you to conclude this

Author Response

R1: Previous observation: meditation is part of karate' tehniques learning, namely through katas' breathing, but this important variable was not analyzed, in either groups.

Authors: That should be the case, but the point is that after surveying Lithuanian karatists, it turned out that far from everyone used this. This study did not aim to determine how the karate technique is learned. However, we see that it makes sense to conduct a separate study in the future.

R1:lines 46-155

Authors: Supplemented.

R1: Materials & Methods 

Authors: Supplemented.

R1: Table 4. Correlations between latent factors and observable exogenous variable- again, identify statistical technique, degrees of freedom, present probability and confidence intervals

lines 267-270- it seems that a linear correlation is not the adequate one, maybe quadratic? inverted U shape? and if so, what the consequences for discussion?

line 277- with so low correlation you need to present confidence intervals, and, once again, you need to identify what statistical technique was used

lines 286-288- identify that they were better in...

lines 288-290- speculative, your experimental design does not allow you to conclude this

lines 299-303- speculative, your experimental design does not allow you to conclude this

Authors: Accepted, added. Data analysis was made with AMOS where aren’t correlations CI conculcated, just p value.  it is possible conculcated CI just for estimates,

Reviewer 2 Report

Dear Authors:
I congratulate you for the work done. I have read it carefully and I liked it very much. I think it is important for the field of Sport Psychology. However, there are some issues that need to be improved so that the work can be accepted for final publication.
1. They describe the characteristics of Kyokushin karate, however it would be appreciated that the practice of Mindfulness would contribute and in which aspects it would improve.
2. The random sampling is complicated to perform, for this reason it should describe in greater depth how the process has been carried out.
3. They should state whether they have permission from the ethics committee and whether the study was conducted in accordance with the Declaration of Helsinki (WMA 2000, Bošnjak 2001, Tyebkhan 2003), which sets out the fundamental ethical principles for research involving human subjects.
5. In the analysis of structural models, in addition to error rates, fit rates and the chi-square/degrees of freedom ratio should be reported. They should also calculate the composite reliability, mean variance extracted, convergent vadilez and validity and discriminant.
6. The main quantitative results should be included in the summary.

I await these modifications.

Yours sincerely.

Author Response

R2:I congratulate you for the work done. I have read it carefully and I liked it very much. I think it is important for the field of Sport Psychology. However, there are some issues that need to be improved so that the work can be accepted for final publication.

A.: Thank you very much.

R2:1. They describe the characteristics of Kyokushin karate, however it would be appreciated that the practice of Mindfulness would contribute and in which aspects it   would improve.

A.: Supplemented.

R2:2. The random sampling is complicated to perform, for this reason it should describe in greater depth how the process has been carried out.

A.: Supplemented.

R2: 3. They should state whether they have permission from the ethics committee and whether the study was conducted in accordance with the Declaration of Helsinki (WMA 2000, Bošnjak 2001, Tyebkhan 2003), which sets out the fundamental ethical principles for research involving human subjects.

A.: Supplemented.

R2:5. In the analysis of structural models, in addition to error rates, fit rates and the chi-square/degrees of freedom ratio should be reported. They should also calculate the composite reliability, mean variance extracted, convergent vadilez and validity and discriminant.

A.: Supplemented.

R2:6. The main quantitative results should be included in the summary.

A.: Supplemented.

Round 2

Reviewer 1 Report

lines 256-7- table 4, probability is repeated

Correlation is not the adequate method to estimate effect size, identify, calculate and present adequate effect sizes

Verify if in CI' signal values are adequately presented, this is, if there are negative values, add negative signal 

Author Response

Reviewer 1: lines 256-7- table 4, probability is repeated

Authors: We have deleted recurring probability.

Reviewer 1: Correlation is not the adequate method to estimate effect size, identify, calculate and present adequate effect sizes

Authors: New calculations are given in Table 4.

Reviewer 1: Verify if in CI' signal values are adequately presented, this is, if there are negative values, add negative signal 

Authors: That table is an AMOS matrix table instead of the interaction of latent (exogenous) data of the same structural model.  As to the scheme, simply the correlation and effect are described. I supplemented the table with additional data, and the text was supplemented  in such manner that the effect became clearer. It is requested to compare differences in the first groups (to apply Mann-Whitney and the like). We did not have such goal; therefore, no such calculations were made, only the structural model of meditating and non-meditating athletes is described. I attach a figure of the structural model of meditating athletes. Should it be included in the article? Since this is not one direction, I cannot perform a path analysis to determine the direct and the indirect effect. I would be grateful for the advice.

Reviewer 2 Report

Dear Authors:
I congratulate you on the work done. I have carefully read your work again. Unless I have read a different version, there are issues that I pointed out in my previous message that have not been resolved

1. You have not included the chi-square ratio/degrees of freedom. Nor have they included composite reliability, mean variance extracted, convergent validity, and discriminant validity.

2. The main quantitative results should be included in the summary.

Yours sincerely.

Author Response

Dear Reviewer,

Thanks for the comments.

Below are your comment and our additions we have made.

Reviewer 2: You have not included the chi-square ratio/degrees of freedom. Nor have they included composite reliability, mean variance extracted, convergent validity, and discriminant validity.

Authors: In Table 2, the chi-square degree of freedom (marked in yellow) is given in parentheses according to APA rules. Composite reliability (CR) and Average Variance Extracted (AVE) are now calculated, they are marked in green and presented in Table 3 in additionally formed two columns. The following sentence is also integrated in the text:

“Composite reliability is more than 0.6 for signs of depression, emotional state and mindfulness, for signs of stress and anxiety it is more than 0.7, so they are a good fit, as Fornell and Larcker (1981) recommended a CR value of 0.60 or more.”

And a new source is integrated:

Fornell, C., & Larcker, D. F. (1981). Evaluating Structural Equation Models with Unobservable Variables and Measurement Error. Journal of Marketing Research18(1), 39–50. https://doi.org/10.2307/3151312). 

Composite reliability (CR)

Average Variance Extracted (AVE)

Emotional state

0,643

0,579

Mindfulness

0,875

0,501

Signs of stress

0,904

0,575

Signs of anxiety

0,784

0,523

Signs of depression

0,682

0,631

Reviewer 2: The main quantitative results should be included in the summary.

Authors: 59.3 per cent of respondents are male, 40.7 per cent are female (N=371). The age of respondents ranged from late twenties to mid-thirties (M=27.75 years, SD=8.56, Min. – 18, Max. – 56). 50.4 per cent have higher university education; 21.8 per cent, secondary; 15.1 per cent, upper secondary; 9.4 per cent, higher non-university; only 1.3 per cent, primary education; and 1.9 per cent, special secondary education. Almost 75 per cent of respondents profess Christianity, 97 per cent of them live in Lithuania. The majority of respondents has practiced kyokushin karate for 11 years and more (53.6 per cent of all respondents), 14.8 per cent of all respondents have practiced this sport for up to 5 years; and 31.5 per cent, for 6-10 years. The more years the athlete has been practicing karate, the higher-ranked belt he or she has (χ² (4) = 240.9, p <.001). 71.4 per cent of karate kyokushin karate athletes who have been practicing this sport for 11 and more years have the 1st dan or a higher belt.